# Downward accountability mechanism effectiveness by non-governmental organizations in low- and middle-income countries: A qualitative systematic review

Elizabeth Noble[1]*, Dina Moinul[1], Oumou Khairy Djim Sylla[1], Sophia Friedmann[1], Kristen Amick[1], Nehal Rowhani[1], Rashi Dua[1], Nowshin Mannan[1], Cathleen Seaman[1], Omobolanle Ayo[1], Shubhra Pant[1], Oluwatimilehin Osoko[1], Srija Gogineni[1], Carly Malburg[1], Chris Dickey[1,2], Emmanuel Peprah[1,3]

**1** Department of Global and Environmental Health, New York University, New York, New York, United States of America, **2** Department of Global and Environmental Health, Applied Global Public Health Initiative (AGPHI), New York University, New York, New York, United States of America, **3** Department of Global and Environmental Health, Implementing Sustainable Evidence-Based Interventions through Engagement Lab (ISEE), New York University, New York, New York, United States of America

* ehn3724@nyu.edu

## Abstract

### Background

Downward accountability, defined as being answerable to beneficiaries for actions and giving affected populations influence in aid processes, remains unstandardized and underinvested across the humanitarian sector. Currently, numerous accountability mechanisms are being utilized by humanitarian non-governmental organizations (NGOs) in low- and middle-income countries (LMICs). However, the different mechanisms have varying degrees of effectiveness in providing true accountability to affected populations due to significant barriers or strengths in implementation.

### Objective

To conduct a qualitative systematic review investigating the various downward accountability mechanisms employed by non-governmental organizations in LMICs, and to assess the effectiveness of these mechanisms in delivering downward accountability for populations in low-resource settings.

### Results

We searched 10 databases, including PubMed, Medline, Embase, Ovid, Web of Science, Global Health, EBSCO SocINDEX, ABI/INFORM, ALNAP, and Sociological Abstracts from 2008–2023. Grey literature was searched on Google Scholar. To capture any additional articles, the search was updated in November 2024. Our search produced 1521 articles. After applying our exclusion criteria and screening,

**Data availability statement:** "All relevant data are within the paper and its Supporting Information files."

**Funding:** The author(s) received no specific funding for this work.

**Competing interests:** The authors have declared that no competing interests exist.

38 articles comprised our final dataset. Each article reported on the effectiveness of five downward accountability mechanisms, including participation, ownership, transparency, program auditing, and social auditing. Associated barriers to accountability included implementation, power asymmetry, and fragmentation within the humanitarian sector.

## Conclusions

There are significant gaps in research on the effectiveness of downward accountability mechanisms amongst humanitarian NGOs in LMICs. This research deficit adversely affects the sustainability of local development initiatives and, on a broader scale, undermines overall organizational effectiveness. Implementing balanced accountability mechanisms that promote equality in power dynamics is pivotal to achieving meaningful outcomes for affected populations.

## Introduction

The significance of downward accountability (DA), defined as being answerable to beneficiaries for actions and giving affected populations influence in humanitarian and international aid, can potentially improve the quality, relevance, and long-term sustainability of activities and initiatives implemented by non-governmental organizations (NGOs) [1]. By fostering genuine engagement with affected communities, DA ensures that NGO activities are better aligned with local needs, thus building trust, promoting transparency, and ultimately increasing the effectiveness and impact of aid interventions [1]. When NGOs strive for immediate results without considering if there will be a sustainable impact, a gap is created between their long-term aspirations and actual outcomes [2]. Establishing credibility with both donors and beneficiaries is vital for securing financial support, trust, and operational success for NGOs. Accountability is crucial in this regard, serving as a fundamental mechanism for ensuring good governance. Lack of accountability has been linked to subpar performance, underscoring its significance in NGO operations [3]. DA is, in effect, an indicator of NGO success. Therefore, a balanced and culturally attuned approach to accountability is essential for NGOs to fulfill their mission of serving their beneficiaries effectively [4].

The neo-institutional theory of accountability examines pressures and power dynamics that shape organizational accountability practices, including within NGOs [5,6]. According to this theory, NGOs encounter various institutional pressures: coercive pressures from donors and regulatory bodies that mandate certain practices, mimetic pressures that drive NGOs to emulate successful organizations, and normative pressures from standards within the sector [6]. NGOs face challenges in balancing the competing responsibilities between upward accountability to donors and DA to beneficiaries [7,8]. These combined forces influence how NGOs structure their accountability mechanisms, ultimately shaping operational and strategic

decisions. As a result, NGOs have historically allocated more resources to meet the demands of donors and regulators at the expense of DA development across the humanitarian sector [9].

Transnationalism has opened opportunities for NGOs to bridge the gap of limitations that both governments and the private sector face in meeting needs [10]. However, the rapid expansion of NGO activities, rooted in globalization and built on colonial foundations, continues to hinder DA [10]. Over the past few decades, the concept of DA in NGOs has gained significant traction due to the growing recognition for NGOs to be held equally answerable to affected populations and their donors [11]. This shift arose from instances where donor-driven priorities, such as during the 2010 Haiti earthquake and the 2014–2016 Ebola outbreak in West Africa, were misaligned with community needs, hindering the effectiveness of aid [11]. In both examples, the emphasis on satisfying donor expectations led to a mismatch in resource allocation and intervention strategies. This ultimately reduced the effectiveness of aid efforts and exacerbated the suffering of those in need.

In response to the misalignment of community needs and donor priorities, the Active Learning Network for Accountability and Performance (ALNAP) emerged as a key research platform in the humanitarian sector [12]. The 2023 ALNAP Report highlights fragmented solutions to political and cultural barriers, emphasizing the need to overcome these implementation obstacles [12]. With the renewed global interest in accountability to affected populations (AAP), initiatives such as the Inter-Agency Standing Committee's 2022 Statement on 'Humanitarian Accountability to Affected People in Humanitarian Action' [13], the Emergency Relief Coordinator's 2023 Statement on AAP [14], and the 2024 Flagship Initiative by the UN Office for Coordination of Humanitarian Affairs [15] all push towards achieving tangible outcomes in DA. Despite these efforts, the Core Humanitarian Standard Alliance 2022 Accountability Report [16] suggests that many NGOs continue to struggle to implement specific commitments to AAP effectively.

Definitions and applications of DA vary across the sector, but it generally involves NGOs engaging communities in planning, evaluation, and delivery of services [17]. DA is a continuous process of engagement, adaptation, and response that aims at achieving sustainable and effective outcomes. However, measuring success in these efforts remains contentious. A greater emphasis on quantitative metrics such as reach and efficiency often overshadows the qualitative aspects of meaningful engagement with affected populations [9].

Various classifications of accountability, such as functional, strategic, formal, and informal mechanisms, reflect its growing importance but also contribute to its fragmentation. Functional accountability ensures task completion and adherence to standards, while strategic accountability focuses on long-term impact [18]. Formal accountability follows official procedures, while informal accountability stems from personal interaction and community engagement [2]. The complexity of defining accountability across the NGO sector is compounded by differing stakeholder expectations and the lack of universally accepted standards, leading to conflicting demands that hinder consistent practices.

Given the critical role that DA plays in ensuring effective aid delivery, the purpose of this systematic review is to explore the various DA mechanisms employed by NGOs in low- and middle-income countries (LMICs). In particular, we seek to assess the effectiveness of these mechanisms in achieving meaningful accountability to affected populations. While much attention has been given to the general concept of accountability, to our knowledge, this is the first systematic review specifically focused on evaluating the effectiveness of DA mechanisms across different contexts in LMICs. By analyzing a range of DA practices and their outcomes, this research aims to identify the strengths and weaknesses of current approaches. Ultimately, our goal is to highlight the practical solutions and strategies that can enhance the implementation of DA, demonstrating its potential to improve the quality and impact of NGO interventions in these regions.

## Methods

A systematic literature review of qualitative case studies on DA mechanisms was conducted according to the Preferred Reporting Items for Systematic Reviews and Meta-Analyses (PRISMA) (S1 Table: PRISMA Checklist) [19]. A systematic review was deemed most appropriate to our research question on the effectiveness of DA mechanisms and to synthesize data

across the humanitarian sector in LMICs. Though qualitative case studies hold a lower strength of evidence, they are the most suitable method for capturing beneficiary experience. For the purpose of this review, accountability tools and processes are referred to broadly as "accountability mechanisms" that often overlap and will be discussed in detail; a tool is used to implement tasks, while a process is a course of action [17]. The review protocol was registered with The Center for Open Science (OSF; https://doi.org/10.17605/OSF.IO/3JNMA). All studies identified in the literature search are available on OSF in accordance with data availability, including eligibility criteria, excluded articles, and data extraction necessary for replicability.

## Search strategy

On 30th December 2023, we searched peer-reviewed literature across 10 databases: PubMed, Medline, Embase, Ovid, Web of Science, Global Health, EBSCO SocINDEX, ABI/INFORM, ALNAP, and Sociological Abstracts. Grey literature was also included and sourced from Google Scholar. Our search was timebound to include only papers published from 2008 to 2023. We rerun the search in November 2024 and determined that no new publications met our inclusion criteria. Three groups of search terms were used, relating to 1) local and international humanitarian NGOs, 2) LMICs, and 3) DA. Synonymous terms for DA were compiled, including beneficiary accountability, accountability to affected populations, humanitarian aid accountability, social accountability, impact evaluation, and answerability. Our full search strategy is included in supplementary materials (S2 Table).

## Eligibility criteria

We assessed the papers against the inclusion criteria (S2 Table). We included qualitative case studies that used data collection methods such as focus groups and interviews with NGO beneficiaries and staff, reporting on DA effect outcomes amongst NGOs in LMICs. Studies that reported on high- and upper-middle-income countries, as defined by the World Bank [20], were excluded along with studies from governmental organizations, those focused on upward accountability, and studies lacking effectiveness outcome data.

## Screening and quality assessment

Title and abstract screening of the initial search was performed independently by all authors using Covidence software [21]. Relevant abstracts underwent a full-text review against inclusion criteria. Two researchers conducted the full-text screening and critical quality appraisal. Two researchers independently used the Joanna Briggs Institute (JBI) Critical Appraisal Tool for Qualitative Research [22] to assess the risk of bias in each of the included studies (S3 Table). This tool is used to appraise the study design, sampling, data analysis, researcher reflexivity, and interpretation of the data. Studies with scores of eight to ten are considered high quality, five to seven medium, and one to four low. To allow for a comprehensive overview, no studies were excluded based on quality assessment. All discrepancies found in the full-text and quality appraisal screening phase were resolved through discussion and by a third researcher.

## Data extraction and analysis

Predefined data was extracted from each study, including study design, methods, findings, and recommendations. Core findings related to DA mechanism effect outcomes and beneficiary experiences were collated into a case study synthesis. The analytic method followed the process of Thomas and Harden (2008), which is derived from the thematic synthesis method that links primary research to implications for practice [23]. Studies were imported to the ATLAS.ti qualitative data analysis software [24] that allows for line-by-line coding used to identify key concepts that are grouped based on similarity and stratified into hierarchical categories. Themes were defined based on a combination of seminal works on DA mechanisms, including frameworks proposed by Edwards and Hulme [7] and the Core Humanitarian Standard Alliance [25], which emphasize accountability to beneficiaries through participatory approaches and transparency. In addition, emergent patterns identified during coding were incorporated. Identified themes were further refined and validated based on their recurrence across multiple studies, particularly those focusing on participation, ownership, transparency, program audits,

and social audits—central components in evaluating NGO accountability mechanisms. Thematic coding performed by two researchers was then independently used to identify interrelationships within the data and maintain consistency across findings. Throughout extraction and data analysis, studies were discussed amongst researchers to reach a consensus.

## Results

The database search yielded 1521 records, with 108 records identified as duplicates and removed. Title and abstract screening were performed on 1413 records, and 107 records were selected for full-text screening. Inclusion and exclusion criteria were applied in full-text screening, resulting in 38 articles deemed eligible for inclusion in this review (Fig 1). A summary of the descriptive characteristics of the 38 selected articles was synthesized (Table 1). Five DA

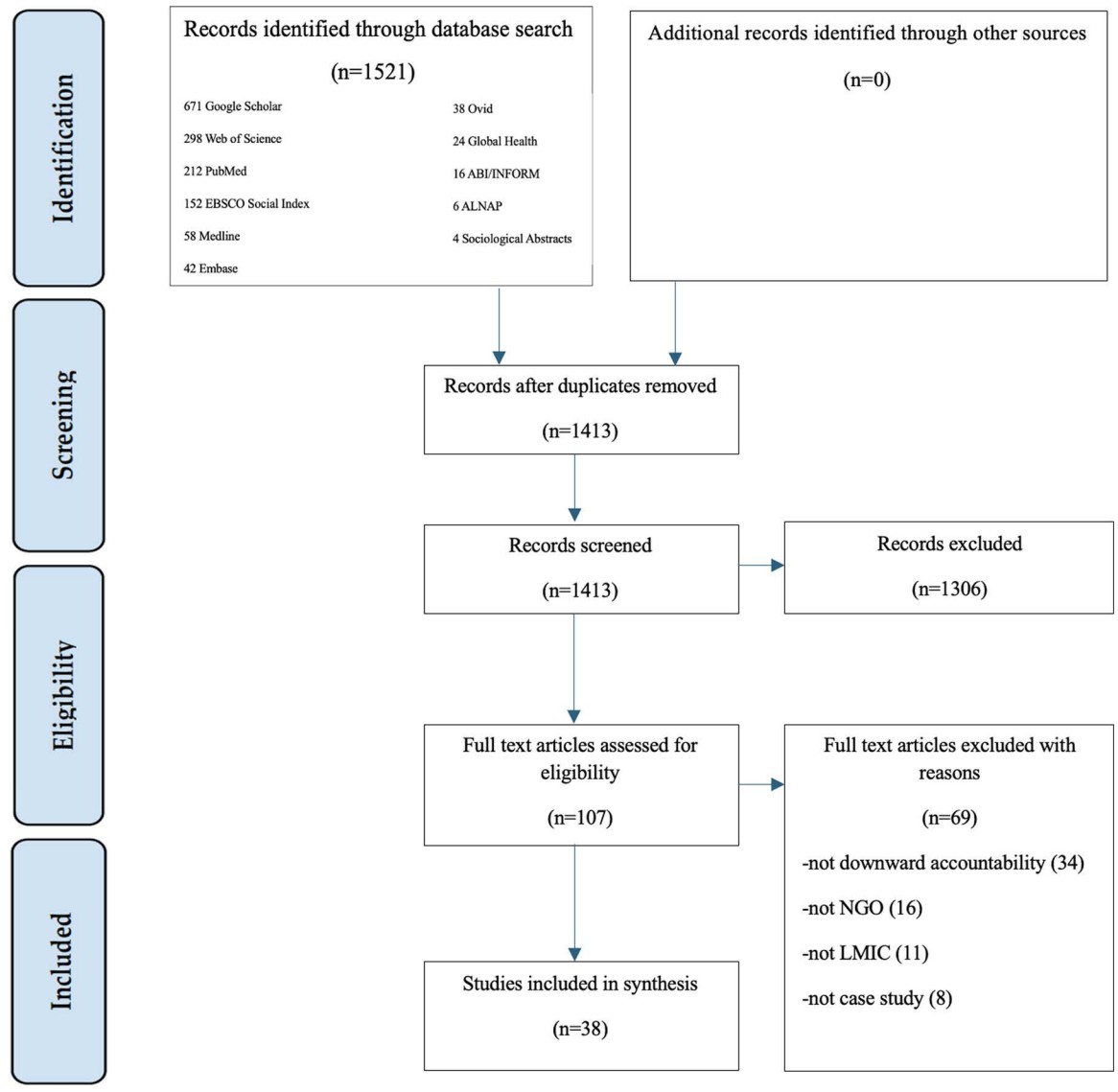

**Fig 1. PRISMA flow diagram.**

**Table 1. Evidence synthesis.**

| Article | Setting | Organiza-tion | DA Mecha-nism | Methods | Results | Recommendations |
|---|---|---|---|---|---|---|
| Abdullah Ahmed, A. et al. (2022) | Yemen | Conglom-erate of 75 NGOs | Program audit | Surveys | Performance assessment and evaluation were found to be the most used practice by the NGOs compared to other practices. Social audit was found to be the least used practice. | Participation and social audits ensure that the community can inclusively monitor processes, ensure NGOs reduce the misuse of resources, and work more effectively in their operations, addressing social barriers and reducing overall resis-tance to interventions. |
| Afsana, K. (2012) | Bangla-desh | BRAC | Partici-pation, ownership | Qualita-tive case study | Participation of community members in health committees empowered them to influence their family members and loved ones to engage in better health practices. | Improving efforts to foster ownership and participation by the local health commit-tee can lead to improved outcomes of participation. |
| Aguiling, M. (2022) | Philip-pines | ACAY Mission Philippines | Feedback system, participa-tion | Surveys, inter-views | Workgroups are participated by more than half of beneficiaries in three NGOs and nearly all beneficiaries in two NGOs. The practice of beneficiary participation is low and trust influences engagement in participation. | Local policy makers, practitioners, and researchers must have an active role to play in ensuring beneficiary participa-tion. Local policy makers need to ensure beneficiary participation is a criterion for NGO accreditation which would increase uptake of DA interventions. |
| Agyemang, G. et al. (2009) | Ghana | Anon-ymous INGO | Transpar-ency | Inter-views | Analyzed the breakdown of promises between donor agencies and recipient governments. | Further research that takes an orga-nizational approach into the nature of unintended consequences deriving from the pressures of upward-accountability requirements could yield results to increase the effectiveness of aid delivery. |
| Amuhaya, D. (2020) | Kenya | Anon-ymous INGO | Program audit, social audit | Qualita-tive case study | Analyzed the existing legal and regulatory frameworks on accountability and pro-ceeded to examine their impact in order to determine the adequacy of the provisions. The inadequacies revolve around the lack of clarity as to the 'what, why, and who' of accountability measures. | The law must make provisions for an integrated approach on how NGOs deal with multiple and competing accountabil-ity demands. The law must ensure that issues to do with accountability are clearly laid out, in alignment with sufficient legal protections, and accountability measures to enable enforcement of practice. |
| Arnott G. et al. (2022) | Uganda | CARE Interna-tional | Feedback system | Qualita-tive case study | The intervention established multiple forums for complaints to be channeled, such as one-on-one confidential consulta-tions, monthly dialogues, home visits, and interactive workshops. These structures facilitated an understanding and claim of human rights. | Integration of international human rights standards within the social accountability process can ensure that marginalized populations and stigmatized issues are not excluded from the accountabil-ity process. It is important to build the capacity and will of health duty-bearers to understand and actively engage with rights-based accountability, coinciding with promoting decentralization of aid. |
| Awio, G., Northcott, D. and Law-rence, S. (2011) | Uganda | Anon-ymous INGO | Participa-tion | Inter-views | Social capital including trust, coopera-tion, and reciprocity, play a crucial role in ensuring effective service delivery and accountability in the humanitarian setting. The program's accountability model, which is driven by social capital and community participation, has led to improved service delivery and positive outcomes. | NGOs should actively involve local com-munities in decision-making and reporting processes to enhance DA. Building and nurturing social capital should be a stra-tegic focus within NGOs to facilitate effec-tive accountability. Simplifying reporting methods to engage a broader community, especially in low-literacy areas, can strengthen DA intervention methods by reducing systemic and practical barriers. |

*(Continued)*

                                    

**Table 1.** (Continued)

| Article | Setting | Organization | DA Mechanism | Methods | Results | Recommendations |
|---------|---------|--------------|--------------|---------|---------|-----------------|
| Awuah-Werekoh, K., Yamoah, F.A., Faizan, R. (2023) | Ghana | Anonymous INGO | Participation | Qualitative case study | There is limited DA to beneficiaries, as donors lack commitment to enforce it. Institutional pressures influence the NGOs' accountability practices. NGOs sometimes circumvent accountability requirements through strategies like acquiescence, manipulation, avoidance, and defiance. | DA is crucial for effective and sustainable development. This gap underscores the need for NGOs to balance their accountability practices to better meet the needs and expectations of their beneficiaries, increasing return on investment by adopting a decolonized approach. |
| Awuku, E. T., Sakyi-Darko, M., & Gyan, M. K. (2020) | Ghana | Anonymous INGO | Participation | Interviews | Community members were less likely to engage in services when the NGO was not transparent about their funding and intentions since they did not know the motivations and felt it was too good to be true, so they feared being trapped or tricked. NGOs are less likely to have DA if there is no external pressure. | Practicing DA by leveraging equitable resource allocation and inclusive systems can help increase the level of trust that host communities have in the organization and help in meeting the organization's objectives. |
| Bawole, J.N., & Langnel, Z. (2016) | Ghana | Anonymous INGO | Participation, social audit | Interviews, focus groups | Even when beneficiaries were involved, the NGO set the rules of behavior for the entire process. The beneficiaries therefore felt they were only being informed, rather than being actively engaged. Even though a feedback system was implemented, and complaints were received, beneficiaries were unable to oversee this accountability tool. | NGOs should engage the existing local political leadership prior to planning the project and reinforce them throughout, to appropriately hold state and non-state actors accountable, centering on rights awareness. |
| Beattie, K. (2011) | South Sudan | Anonymous INGO | Social audit, participation, transparency | Interviews, focus groups | There were improvements to DA measures when there were staff designated to it. There were many contradictions between what the mechanisms were meant to do, what expectations they relied upon, and what the community members did and found more important. Even when the mechanisms were implemented the community members went a different route to be heard. | Stakeholder accountability analyses will inform staff and community, identify conflicts, and showcase connections and relationships. Dedicated staff that are relationship-oriented and are liaisons with the community can be helpful in discovering accountability mechanisms that align with community values. |
| Brunt, C. and McCourt, W. (2012) | Kenya | 7 INGOs | Participation, social audit | Interviews | Beneficiaries reported that their participation in regular meetings and workshops ensured successful program delivery and improved communication with donors. INGOs did not generally follow through with what they said they would do. | Long-term impact to beneficiaries should be the focus of NGOs, while pressure from donors for rapid results can hinder this progress and neglect underlying systemic issues. When practical inhibitory factors and sectoral fragmentation are not adequately assessed, inequalities are exacerbated and catalyze failure in implementation strategies in the community. |
| Burger, R. Owens, T. (2010) | Uganda | 300 Ugandan NGOs | Evaluation reporting | Surveys | The current reliance on officially reported information for regulating and monitoring NGOs may not be sufficient. The findings suggest that conflicts with the government play a significant role in misrepresentation. | There is a growing need for reputable, trustworthy third-party monitoring to aid in the development of efficient and effective accountability interventions. |

*(Continued)*

**Table 1.** (Continued)

| Article | Setting | Organiza-tion | DA Mecha-nism | Methods | Results | Recommendations |
|---|---|---|---|---|---|---|
| Chu, V. T. T. (2015) | Bangla-deshIn-donesia | Anon-ymous INGO | Participa-tion | Inter-views | NGO accountability to beneficiaries focused on a socializing dimension by engaging beneficiaries, the private sector, and local government in programs while emphasizing the active role of beneficia-ries, facilitating a participatory approach to accountability. | Mechanisms for NGO accountability to beneficiaries can remain informal but should be easy to use and accessible for both NGOs and beneficiaries, enabling effective exchange of information, interac-tion, and facilitating dialogue–with partici-pation underpinning these mechanisms. |
| Chu, V., & Luke, B. (2018) | Vietnam | Vien Chu | Participa-tion | Inter-views | The study revealed a three-level frame-work for effective beneficiary participation involving consultation, partnership, and delegated control. This framework pro-vided a graduated approach to empower-ing beneficiaries, and a basis for enhanced accountability of NGOs at both a functional and a strategic level. | A deliberate strategy of making spaces for meaningful participation by a range of actors and stakeholders emphasizes a power which NGOs should embrace, and donors should enforce. Assigning control at a functional level, while also maintain-ing control at a strategic level, is a difficult balance and can be attainable when enhanced communication and transpar-ency are prioritized. |
| Desie, S., & Ismail, M. O. (2017) | Somalia | World Food Pro-gramme | Social audit, program audit | Qualita-tive case study | Program evaluations can identify where beneficiary accountability needs improvement and how implementing new accountability mechanisms can better ensure quality services are delivered to the intended beneficiaries. | Development of an accountability to affected populations (AAP) framework improves implementation of AAP mecha-nisms by providing a structured approach and strategy between monitoring and evaluation practices for NGOs and beneficiaries. This would align service aid delivery with community needs. |
| Dewi, M. K., Manochin, M., & Belal, A. (2019) | Indonesia | ACT | Participa-tion | Qualita-tive case study | Engaging local volunteers from beneficia-ries' proximity improves NGO account-ability. The social accountability approach works well for localized beneficiary accountability. | Developing training programs to equip volunteers with the necessary skills and knowledge can lead to effective engagement. Fostering an environment that encourages volunteers to contribute their social and cultural capital strength-ens community voices and promotes collective action. When NGOs encourage interdependent decision-making between volunteers and beneficiaries, they empower the latter. |
| Dewi, M. K., Manochin, M., & Belal, A. (2021) | Indonesia | Anon-ymous INGO | Com-munity surveil-lance | Inter-views, focus groups | DA that is casually demanded and advo-cates against excessive formal report-ing facilitates effective communication between beneficiaries and NGOs. DA that is action-based emphasizes the impor-tance of beneficiaries receiving tangible assistance as a result of program delivery, thereby fostering a sense of accountabil-ity from the implementing organization, transparency in fund allocation, ensuring accountability to the community where programs are implemented. DA prioritizes beneficiary self-reliance. | NGOs can cultivate sustainable pro-grams by involving beneficiaries from the planning phase onward. It is essential for beneficiaries to play a central role in conducting effective needs assessments and justifying the design and delivery of services. Active participation in NGO pro-grams promotes stronger bonds between beneficiaries and NGOs, enhancing coop-eration, and ultimately leading to program success and sustainability. |

*(Continued)*

**Table 1.** (Continued)

| Article | Setting | Organization | DA Mechanism | Methods | Results | Recommendations |
|---|---|---|---|---|---|---|
| Jacobs, A., & Wilford, R. (2010) | Angola | Anonymous INGO | Program audit, participation, evaluation reporting | Focus groups | Self-assessment was constructive to provide examples of the importance of DA, to dedicate time and energy to creating methods to improve DA and reiterate the mission of helping a community become self-sustaining. Community research was successful in determining the communities' opinions on how the NGO was doing in relation to effective DA delivery. | Future progress will depend on senior decision makers creating incentives for managers to focus on DA and monitoring actual performance. Given the context of packed management agendas and often overworked field staff, management priorities may have to be carefully reviewed to create the space and support for staff to deliver what matters most, emphasizing community-led discussions that increase uptake across local and national levels. |
| Kirsch, D. C. (2013) | Pakistan | Anonymous INGO | Feedback system | Interviews, surveys | Complaint processes are the least used accountability mechanism, while learning is most frequently not written. The use of accountability mechanisms in general, and written accountability mechanisms specifically, tends to increase with NGO size. | It is advisable for NGOs to develop the tools and indicators that will determine the effects of their accountability mechanisms, so that they can use this information to inform their actions and adapt accountability practices based on feedback. |
| Komujuni, S. Mullard, S. (2020) | Uganda | UNHCR | Feedback system, program audit | Interviews | DA does not always yield clear benefits, and there is no universally effective design; each mechanism presents its own challenges. UNHCR faces significant financial losses annually due to incidents of mismanagement, corruption, and fraud within its refugee protection system. | Recommendations consist of employing participatory methods in creating DA mechanisms, enhancing communication channels, and engaging beneficiaries throughout the planning, design, and implementation phases. Requiring DA training as a condition for donor funding are essential measures in participatory engagement. |
| Leigh, J. A. (2019) | Ghana | Anonymous INGO | Ownership | Qualitative case study | Large NGOs have more direct oversight than many donors, and more easily earn the trust of implementing partners. | Harness donor engagement and shift power to affected populations through direct funding to local organizations. |
| Lloyd, R., et al. (2008) | Pakistan | Centre for Philanthropy | Program audit | Qualitative case study | Lacking definition of DA makes implementation and enforcement challenging due to the lack of standardization and the inability to evaluate or compare efforts. | Clarifying what DA means, within historical frameworks that inform current decisions, would help create standardized guidelines on how it is to be evaluated and implemented. |
| Manilla Arroyo, D. (2014) | Haiti | IFRC, Red Crescent Societies, and Oxfam Great Britain | Participation, evaluation reporting, social audit | Qualitative case study | 87% of people surveyed had received information from the IFRC and 88% of those said the information was useful. Conversely, Oxfam GB implemented a similar system for complaints and only 20% of calls were actual complaints. People voiced that their needs had changed from emergency assistance to livelihood and long-term housing, outside of NGO scope. The overall result was low utilization of complaint response mechanisms and limited response to feedback concerns. | Developing technologies is a promising avenue of communication between NGOs and affected populations. However, information mechanisms must not only communicate what an NGO provides, but also what one is entitled to receive in basic human rights. Phone calls should not be assessed on a participatory level when their main function is for feedback systems. Participation was found difficult to measure and thus, the mechanism for which less impact evaluation is available. |

*(Continued)*

**Table 1.** (Continued)

| Article | Setting | Organization | DA Mechanism | Methods | Results | Recommendations |
|---|---|---|---|---|---|---|
| Massud, M. E. I., & Aktar, A. (2020) | Bangladesh | Ain O Shalish Kendr (ASK) | Ownership | Qualitative case study | ASK found that utilizing social capital is effective in discharging accountability towards beneficiaries by maintaining a partnership network of community experts of different fields of human rights, utilizing their expertise in the process of DA delivery. They also used media to ensure transparency in communicating program activities. | Social capital contributes to the achievement of beneficiary accountability but does not ensure complete functionality of beneficiary accountability. This is mainly for the lack of ownership type control by the beneficiaries. Future studies must use more data collection tools. |
| Mawanda, H. N. (2012) | Kenya | Anonymous local NGOs | Social audit | Surveys | Accountability to community stakeholders was found to be quite low. Social audits were found useful towards governments and donor agencies to ensure effective DA delivery. | NGOs and key leaders that adopt internal policies that facilitate, and champion DA practices yield on investment by establishing a pro-accountability culture. |
| Mercelis, F., Wellens, L., & Jegers, M. (2016) | Vietnam | Vredeseilande | Participation | Qualitative case study | Only half of beneficiaries reported being explicitly asked for feedback on NGO activities. Feedback did not focus on governance or broader organizational strategies. A prominent level of beneficiary participation was observed, but not involvement in strategic decision-making. This is found to be lacking in partnership and control, instead it remains at a weak participation level. | Acknowledging the significance of informal and secure communication channels can foster dialogue between beneficiaries and their representatives. Alleviating resource constraints by providing training in negotiation skills would cultivate an environment of equal partnership. |
| Mir, M., & Bala, S. K. (2015) | Bangladesh | Anonymous INGOs | Program audit | Qualitative case study | INGOs with internal funding align programs with community needs, enhancing DA. INGOs publish reports in newspapers and online, improving public and beneficiary access. INGOs involve beneficiary community groups due to financial independence, emphasizing a participatory approach. | The following interventions can amplify collective community commitment and reduce sectoral fragmentation: establishing independent oversight bodies to diminish political influence, developing standardized reporting frameworks to enhance transparency and benchmark NGO performance, and implementing mandatory performance audits for NGOs that make service outcomes public, and by providing incentives like funding and recognition for follow-through. |
| O'Leary, S. (2017) | India | Rural Life and Unison | Feedback system | Qualitative case study | Understandings of transformative learning between NGOs differed, often depending on the core focus of the NGO. Participatory accountability mechanisms are found redundant when they are misaligned with the realities of beneficiary communities, particularly when they neglect the capabilities that beneficiaries possess, and the power relations that shape them. | In order to critically appraise accountability practice, the underlying motivations and potentially transformative intentions of the actors involved need to be understood, particularly in terms of how these contribute to and influence the overriding objectives embodied within certain accountability relationships and the accounting practices that facilitate these promises. |

*(Continued)*

**Table 1.** (Continued)

| Article | Setting | Organization | DA Mechanism | Methods | Results | Recommendations |
|---|---|---|---|---|---|---|
| Price, R. (2018) | Philippines Haiti | Anonymous INGO | Feedback system | Qualitative case study | There are a variety of feedback mechanisms and communication channels; face-to-face is often the preferred method. Contextual sensitization and two-way communication were found more beneficial, instead of only extracting information. | Feedback systems require increasing accessibility, ease of use, timely response to close the loop, and participation in future planning designs while empowering communities and altering traditional power structures. |
| Rahman, M. (2014) | Bangladesh | Anonymous INGO | Program audit | Qualitative case study | Donor influence and the nature of governing bodies significantly impact NGO accountability. NGOs share information through various means, with donors receiving more detailed data - detracting from transparency. Beneficiary participation is limited, with NGOs often guiding discussions. Participation is found more symbolic than substantive. | Improve resources and DA mechanisms of NGOs for better regulation. Rethink internal governance structures for better accountability and transparency. Create more meaningful engagement opportunities. |
| Schaaf, M., Topp, S. M., & Ngulube, M. (2017) | Zambia | World Vision | Capacity building, knowledge transfer | Qualitative case study | Healthcare workers exhibited improved behavior, leading to better treatment, reduced absenteeism, and increased access to officials, with communities now expecting accountability. This strengthened interactions between state and society, enhancing transparency in meetings and promoting deeper engagement between citizens and leaders, further promoting social accounting. | The study underscores the efficacy of implementing the social accountability framework that increased community awareness of rights and improved coordination of service delivery between NGOs and governmental bodies. Implementing DA mechanisms requires proper funding to establish adequate resource allocation, and capacity-building to ensure long-term impact. |
| Siddiquee, N. A., & Faroqi, M. G. (2009) | Bangladesh | Anonymous INGO | Feedback system | Qualitative case study | A variety of formal accountability mechanisms are found lacking in effective DA delivery. The lack of regulatory frameworks across the NGO sector further inhibits DA. | There is an urgent need for various strategies and mechanisms of NGO accountability to be revisited and recalibrated. When affected populations have less influence in NGO activities, it is important to establish greater supervision which can reinforce effective feedback and follow-up mechanisms. |
| The Humanitarian Accountability Partnership International (2010) | Haiti Pakistan | HAP Intl. | Feedback system | Focus groups, interviews | Limited information-sharing with beneficiaries leads to frustration and confusion. Community consultations and complaint mechanisms were found essential in improving perceptions of aid agencies. Certification against standards like HAP on organizational performance, with the growing role of information technology, is impactful towards effective DA delivery. | Implement coordinated information dissemination strategies to ensure that beneficiaries are well-informed about aid programs, processes, and opportunities for feedback and complaints. Recognize the importance of allocating adequate time and resources to accountability processes, viewing them as integral components to humanitarian operations and in turn, strengthening organizational practices. |
| Uddin, M. M., & Belal, A. R. (2019) | Bangladesh | BRAC | Community surveillance | Qualitative case study | NGOs may use participatory tools primarily for legitimization purposes. The evidence demonstrates that beneficiary participation in BRAC's donor-funded programs facilitates accountability through dialogue. These dialogues lead to active involvement of beneficiaries in local decision-making processes. | Donors can play a proactive role in promoting accountability amongst NGOs, especially when beneficiaries lack the power to hold NGOs accountable. Long-term studies may reveal how change in donor requirements impact beneficiary accountability over time, and measure sustainability when stronger governance practices are implemented. |

*(Continued)*

**Table 1.** (Continued)

| Article | Setting | Organization | DA Mechanism | Methods | Results | Recommendations |
|---|---|---|---|---|---|---|
| Uddin, M.M., & Belal, A.R. (2014) | Bangladesh | Anonymous INGO | Ownership, transparency | Qualitative case study | The donor-funded program implemented more DA mechanisms, resulting in better program effectiveness and capacity building. Both were used as benchmarks for DA delivery. | Use donor accountability to strengthen DA. As skill development and evaluation usually are elements of accountability - there can also be participation written in as it allows for more structured implementation, measurement, and enforcement. |
| Wardwell, S. E. (2012) | Haiti | Anonymous INGO | Social audit | Surveys | Analyzed effects of social auditing on DA delivery. | Traditional approaches to DA are insufficient and must be improved upon because current approaches fail to fully deliver beneficiary accountability. A commitment to alternative social approaches of DA interventions would improve stakeholder engagement and foster better decision-making. |
| Walsh, S. (2016) | Uganda | ActionAid | Participation | Qualitative case study | Although ActionAid established forums for consultation and transparency to incorporate community input into decisions, the actual implementation of these mechanisms did not align well with their intended purpose, demonstrating one-sided dialogue and lacking cultural awareness. | NGO staff may lack the necessary skills in community engagement, interpersonal communication, language proficiency, and values essential for connecting with the communities served. Centralizing the dissemination process of accountability mechanisms may ensure equal emphasis on both upward and downward accountability. |

mechanism groups were identified for effect outcome analysis, including participation, ownership, transparency, program auditing, and social auditing (Table 2). Qualitative analysis led to the creation of final themes and concepts (Table 3). This framework enabled the authors to compile the findings presented in this review, which includes a comparative analysis of DA frameworks, the effectiveness of DA mechanisms, as well as their strengths and barriers to effective implementation.

**Table 2.** DA mechanism group definitions.

| DA Mechanism | | Definitions | References |
|---|---|---|---|
| Participation | Process | Community engagement, consulting, involving, collaborating | [17,26,29,32] |
| Ownership | Process | Knowledge transfer, capacity building, sustainability for self-reliance, integration with existing systems, local leadership, beneficiaries making the decisions and those decisions are implemented | [9,12,17,31] |
| Transparency | Mixed Tool-Process | Evaluation reporting; Disseminating information to communities on how finances are used, organizational structure, and program outcomes; Communicating motives, resources, power dynamics, limitations of work, and decision-making process; Aim to maintain community trust, being upfront about access to human rights | [9,10,12,17] |
| Program Audit | Tool | Internal assessment, performance evaluation, reporting and improving on social performance and ethical behavior, self-regulation and enforcement | [17,34,39,42] |
| Social Audit | Mixed Tool-Process | Community surveillance, needs assessment, focus groups, surveys; Feedback systems, complaint mechanisms, participatory evaluation | [10,17,42,43] |

**Table 3. Thematic coding results.**

| Thematic Code | Description | Files | References |
|---|---|---|---|
| **1. History of accountability** | The evolution and past practices of accountability mechanisms in the sector. | 13 | 34 |
| 1.1. Difficulty defining across the sector | Challenges in establishing a unified definition of accountability within the sector. | 11 | 32 |
| 1.2. Unequal allocation of resources towards upward accountability | Disproportionate focus and resources directed at accountability to donors and authorities rather than beneficiaries. | 13 | 23 |
| 1.3. NGO frameworks used in the field | Various frameworks currently adopted by NGOs to implement accountability. | 11 | 36 |
| 1.4. Need for a decolonized approach to NGO accountability mechanisms in global health | The necessity to adopt accountability mechanisms that are free from colonial influences in global health. | 6 | 20 |
| 1.4.1. Importance of accountability | The critical role of accountability in ensuring transparency, trust, and effectiveness within the sector. | 6 | 13 |
| **2. Feedback systems** | The process and practice of gathering and addressing input and grievances from stakeholders. | 2 | 3 |
| 2.1. Mechanism effects | The impact outcomes of implementing accountability mechanisms. | 7 | 16 |
| 2.2. Strength/Weakness in implementation | The advantages and shortcomings observed during the application of accountability mechanisms. | 18 | 41 |
| 2.2.1. Failure to act | Instances where accountability mechanisms were not activated or followed through. | 6 | 12 |
| 2.2.2. Poor usage | Ineffective utilization of mechanisms due to fear of losing aid, reluctance to complain, or lack of response. | 7 | 12 |
| 2.2.3. Mistranslation | Feedback mechanisms failing to communicate effectively or fulfill community expectations. | 2 | 2 |
| **3. Capacity building** | The process of sharing knowledge and enhancing skills and abilities within the sector. | 0 | 0 |
| 3.1. Mechanism effects | The impact outcomes of knowledge transfer and capacity building efforts. | 7 | 10 |
| 3.2. Strength/Weakness in implementation | The advantages and shortcomings observed during the execution of knowledge transfer and capacity-building initiatives. | 4 | 4 |
| **4. Ownership** | The process of enabling stakeholders to make the decisions and implement them. | 5 | 6 |
| 4.1. Mechanism effects | The impact outcomes of ownership and empowerment initiatives. | 15 | 28 |
| 4.2. Strength/Weakness in implementation | The advantages and shortcomings observed during the execution of ownership and empowerment efforts. | 12 | 21 |
| **5. Participation** | The involvement of stakeholders in accountability processes and decision-making. | 5 | 7 |
| 5.1. Mechanism effects | The impact outcomes of participatory mechanisms. | 19 | 59 |
| 5.2. Strength/Weakness in implementation | The advantages and shortcomings observed during the application of participatory mechanisms. | 5 | 31 |
| 5.3. Difficulty defining scope of mechanism | Challenges in clearly delineating the extent and boundaries of participatory mechanisms. | 17 | 58 |
| **6. Social audit** | The practice of community-led monitoring and evaluation of accountability processes. | 3 | 4 |
| 6.1. Mechanism effects | The impact outcomes of community surveillance and social auditing efforts. | 9 | 13 |
| 6.2. Strength/Weakness in implementation | The advantages and shortcomings observed during the application of community surveillance and social auditing. | 8 | 11 |
| **7. Program audit** | The systematic evaluation of programs to ensure compliance and effectiveness. | 4 | 7 |
| 7.1. Mechanism effects | The impact outcomes of program auditing efforts. | 8 | 16 |

*(Continued)*

| Thematic Code | Description | Files | Refer-ences |
|---|---|---|---|
| 7.2. Strength/Weakness in implementation | The advantages and shortcomings observed during the execution of program auditing processes. | 11 | 23 |
| **8. Evaluation reporting** | The process of documenting and communicating the results of evaluations. | 1 | 1 |
| 8.1. Mechanism effects | The impact outcomes of evaluation reporting efforts. | 2 | 2 |
| 8.2. Strength/Weakness in implementation | The advantages and shortcomings observed during the execution of evaluation reporting processes. | 3 | 3 |
| **9. Transparency** | The practice of providing clear and comprehensive information to stakeholders. | 8 | 8 |
| 9.1. Mechanism effects | The impact outcomes of informing and transparency efforts. | 11 | 20 |
| 9.1.1. Financial transparency | The clarity and openness in financial disclosures and reporting. | 1 | 2 |
| 9.2. Strength/Weakness in implementation | The advantages and shortcomings observed during the application of informing and transparency mechanisms. | 13 | 27 |
| 9.2.1. Misrepresentation to maintain reputation | Instances where information is distorted or misrepresented to uphold organizational reputation. | 2 | 7 |
| 9.2.2. Miscommunication | Problems arising from ineffective or erroneous communication practices. | 3 | 3 |
| **10. Barriers To accountability** | Factors hindering the effective implementation of accountability mechanisms. | 0 | 0 |
| 10.1. Implementation | Challenges related to the execution of accountability mechanisms. | 14 | 32 |
| 10.1.1. NGO size in relation to governance practice | Influence of NGO size on governance practices. | 2 | 2 |
| 10.1.2. Lack of organizational staff training | Insufficient training of staff members within organizations. | 6 | 14 |
| 10.1.3. Lack of resources | Inadequate allocation of resources. | 8 | 10 |
| 10.1.4. Lack of impact monitoring | Absence of tools or systems to monitor and evaluate impacts. | 2 | 2 |
| 10.1.5. Follow-up mechanisms | Inadequate procedures for follow-up and enforcement. | 1 | 1 |
| 10.1.6. Conflicting goals | Divergent objectives among stakeholders. | 8 | 10 |
| 10.1.7. Action planning for local development | Deficiencies in planning and executing local development initiatives. | 2 | 5 |
| 10.1.8. Cultural/Social | Cultural and social factors influencing accountability practices. | 3 | 14 |
| 10.2. Power asymmetry | Imbalances in power dynamics affecting accountability. | 10 | 23 |
| 10.2.1. Structural systemic challenges | Systemic issues undermining accountability structures. | 6 | 9 |
| 10.2.2. Competing priorities | Conflicting interests and priorities. | 10 | 24 |
| 10.2.3. Perceived beneficiary capability | Assumptions about the capacity of beneficiaries to engage in accountability processes. | 7 | 14 |
| 10.3. Fragmentation within the humanitarian sector | Disintegration and disunity within the humanitarian field. | 3 | 6 |
| 10.3.1. Lack of national level engagement from grassroots movement | Insufficient involvement of grassroots movements at the national level. | 5 | 8 |
| 10.3.2. Lack of sectoral governance structures and institutional strengthening | Weak governance structures and institutional capacity. | 8 | 14 |
| 10.3.3. Ability of NGOs to not consider themselves responsible for an action | NGO evasion of responsibility for actions. | 5 | 11 |
| **11. Future directions** | Prospective avenues and areas for further exploration in account-ability practices. | 0 | 0 |
| 11.1. Research gaps | Areas in accountability research requiring further investigation. | 12 | 19 |
| 11.2. Decolonizing humanitarian aid in the context of local development | Efforts to dismantle colonial influences in humanitarian aid. | 2 | 5 |
| 11.2.1. Historical effects of colonization on power dynamics | Understanding the impact of colonization on power structures. | 4 | 5 |

*(Continued)*

**Table 3.** (Continued)

| Thematic Code | Description | Files | References |
|---|---|---|---|
| 11.2.2. Terminology of "beneficiary" | Critique of the term "beneficiary" and its implications for perpetuating inequality. | 2 | 2 |
| 11.2.3. The concept of DA as inherent to power asymmetry | Exploration of the inherent power dynamics in DA. | 2 | 2 |
| 11.2.4. Organizational legitimacy | Examination of the legitimacy of DA when measures are set by organizations. | 2 | 5 |
| 11.2.5. Social justice | Consideration of social justice principles in humanitarian aid practices. | 3 | 3 |
| 11.2.6. Ownership equates sustainability | Linking ownership to sustainability in development initiatives. | 2 | 3 |
| 11.3. Integration with existing systems | Incorporating accountability mechanisms into pre-existing structures. | 7 | 10 |
| 11.4. Facilitators of implementation | Strategies to enhance the effectiveness of accountability mechanisms. | 7 | 10 |
| 11.4.1. Prioritization of local development systematically | Systematic prioritization of local development initiatives. | 6 | 7 |
| 11.4.2. Administrative structures | Enhancing management and administrative frameworks. | 5 | 13 |
| 11.4.3. Effectiveness measures and information system management | Developing measures of effectiveness and managing information systems. | 3 | 5 |
| 11.4.4. Incorporate DA as a measure of program effectiveness | Integrating DA as a measure of program effectiveness. | 2 | 2 |
| 11.4.5. Standard regulatory framework | Establishing a standardized regulatory framework for accountability. | 5 | 11 |
| 11.4.6. Leveraging donors and upward accountability for DA | Utilizing donors and upward accountability mechanisms for DA. | 8 | 10 |
| 11.4.7. External pressures | Utilizing external pressures from civil society, media, and local governing bodies to enhance accountability. | 5 | 10 |
| 11.4.8. Allocation of resources towards DA | Allocating resources towards DA initiatives. | 3 | 10 |
| 11.4.9. Financial independence | Achieving financial independence to enhance accountability for future initiatives. | 1 | 3 |
| 11.4.10. From functional to social accountability | Transitioning from functional to social accountability and addressing legitimacy concerns in DA. | 2 | 13 |
| 11.4.11. Staff attitudes | Addressing staff attitudes towards accountability mechanisms. | 3 | 9 |
| 11.4.12. Clear stakeholder analysis | Conducting clear stakeholder analysis to inform accountability practices. | 1 | 1 |

## Accountability frameworks

We synthesized a comparative analysis of the core components, advantages, and limitations of accountability frameworks. This analysis provides a foundation for understanding how these frameworks influence NGO performance outcomes (Table 4).

## Participation

The mechanism of participation is often used synonymously with accountability as it transcends across many processes and tools (Table 3). We found that effective participation mechanisms in NGO projects, such as those implemented in Haiti's poverty alleviation and microenterprise development programs, are crucial for achieving active beneficiary involvement [26]. Engagement with beneficiaries through participation enhances economic and social outcomes for the underserved by involving them in business ventures, decision-making processes, and community dialogue. This engagement

**Table 4. Accountability frameworks.**

| | |
|---|---|
| Accountability, Learning and Planning System (ALPS) | Introduced by ActionAid in 2000, the ALPS framework is designed to embed DA into the heart of organizational procedures [8]. ALPS prioritizes internal coordination and aims to transfer power from donors to beneficiaries by focusing on long-term accountability rather than short-term outcomes. The framework advocates for less reporting to central offices and more direct communication with local communities, involving them in planning and evaluation processes, and utilizing donor interactions to enhance accountability. |
| The Active Learning Network for Accountability and Performance in Humanitarian Action (ALNAP) | ALNAP enhances learning and accountability in humanitarian efforts [12]. The State of the Humanitarian System Report, produced by ALNAP regularly, identifies both advancements and deficiencies in humanitarian accountability. |
| Core Humanitarian Standard (CHS) | The CHS outlines nine commitments for humanitarian organizations to enhance the quality, relevance, and effectiveness of aid [25]. It addresses accountability, coordination, learning, people management, prevention of exploitation, abuse, fraud, and corruption. |
| Do No Harm | The Do No Harm framework, initially created to evaluate the effects of aid on conflict scenarios, underscores accountability via a feedback loop. Acknowledging its shortcomings in community involvement, the framework has been refined through initiatives such as the Listen Project, which incorporates the perspectives of local populations more effectively [10]. |
| Good Enough Guide | The Good Enough Guide establishes minimum standards for NGO activities during emergencies, emphasizing transparency, participation, and feedback [10]. Although these guidelines are voluntary and lack stringent enforcement mechanisms, they offer valuable insights that can enhance long-term development initiatives. |
| The Grand Bargain | Established in 2016 at the World Humanitarian Summit, the Grand Bargain put forth commitments to improving aid effectiveness with an annual review of progress [56]. The focus was on donor-driven accountability that improves resource management for greater responsiveness to affected populations. While this initiative has seen great progress in policy change for localization, there is much room to improve systemic practice outcomes for demand-driven aid [56]. |
| Humanitarian Accountability Partnership (HAP) | Founded in 2003, HAP set forth standards for humanitarian accountability and quality management, emphasizing accountability to beneficiaries [8]. The HAP Standard delineates six performance benchmarks: setting commitments, ensuring staff competency, sharing information, encouraging participation, handling complaints, and forwarding continuous improvement [38]. Certification under HAP ensures adherence to these standards [11]. |
| Listen First | Developed by Concern and Mango, this framework offers a pragmatic approach to defining DA, establishing performance standards in four key areas: dissemination of public information, stakeholder participation in decision-making, responsiveness to stakeholders, and staff attitudes and behaviors [8]. These areas are evaluated on a four-point scale categorized as sapling, maturing, flowering, and fruit-bearing. Each category includes illustrative behaviors, with a focus on gender and power dynamics, to support and guide staff in their engagement with marginalized communities. |

not only promotes self-sufficiency but also ensures that projects are tailored to meet local needs. Collaboration with local authorities and the community to integrate local knowledge was identified as a critical factor [26]. By facilitating two-way communication, participatory representation allows beneficiaries to express their needs and opinions directly, enhancing the effectiveness of NGO activities via focus group discussions and community consultations [27,28]. However, our review also noted that while digital technologies for feedback are beneficial, they often create an echo chamber effect, which can limit meaningful dialogue with NGOs [29].

A common theme throughout our review is that implementing participation mechanisms presents a complex landscape of strengths and weaknesses. While there is a stated intention to empower communities, participation often remains superficial, characterized more by consultative processes than truly collaborative engagement [30]. Our assessment reveals that several factors contribute to this dynamic, including a paternalistic approach by humanitarian staff and inadequate evaluation of accessible communication tools that hinder meaningful engagement in humanitarian contexts [29]. Many participants perceive NGO initiatives as externally imposed rather than genuinely inclusive, undermining sustainable DA [31].

The literature suggests that the facilitators for DA include effective participation that requires cultivating trust, establishing shared goals, and facilitating effective communication for collaboration between NGOs and affected populations [32]. We found that formal meetings leave communities feeling excluded from policymaking and fund allocation

decisions [32]. Thematic barriers to effective participation include insufficient staff training and resources, such as inadequate budget allocation, which limit the successful implementation of participation initiatives [33]. Additionally, there is a lack of effective impact monitoring and follow-up mechanisms, exacerbating implementation challenges [18]. Furthermore, conflicting organizational priorities and commitments complicate participation delivery, as NGOs can prioritize their own agenda over community interests [30]. An emerging theme across various studies was that power imbalances within organizational structures play a significant role when donor priorities overshadow local community voices [30,33,34]. The level of perceived community capability influences involvement in decision-making processes [9]. Limited community awareness of their rights contributes to sustaining this power imbalance and reduces the demand for NGOs to be held accountable [12,35].

## Ownership

Ownership emerged as a significant theme in our qualitative systematic review of the humanitarian sector (Table 3). Ownership signifies a deliberate act of empowerment through tools such as knowledge transfer and capacity building, aiming for self-reliance. Ownership and empowerment are frequently equated with accountability or seen as outcomes of DA [9,31,36]. We found that ownership is implemented through direct financial assistance and micro-enterprise interventions in several studies [9,31,36]. These interventions provided technical support while allowing beneficiaries to make financial decisions, shifting power to beneficiaries and placing them at the center of program design and evaluation [9,31,36]. Capacity building and knowledge transfer are effective delivery methods for DA that inherently demand empowerment, by connecting community members' capacity and knowledge to their goals [9].

An aspect of ownership is the transfer of resources, in the form of information and skills, that validates DA and necessitates community buy-in [37]. Various studies demonstrate that this mechanism allows community members to be agents of change, creating demand and fulfilling service roles traditionally managed by NGOs [9,31,36]. Empowered beneficiaries can appraise the NGO using success criteria they devise themselves [37]. Other mechanisms such as participation, feedback, transparency, and social audits play significant roles in community empowerment and ownership in delivering DA.

## Transparency

Current practices identified as necessary for DA include transparency and information-sharing between the disclosure of evaluation reports to upward stakeholders, often requiring legal documentation in the case of financial data [10,34], whereas advocates for DA convey the process of relaying this same information to communities as equally essential (Table 3). This overarching theme indicated that the act of informing does not translate directly to effective delivery of DA unless it is utilized as a tool to advance credibility and trust, resulting in increased community participation [6,35].

For example, the type of information that is shared with beneficiaries includes project aims, NGO activity progress, and the prospect of adding value to the community [38]. However, information regarding financial statements and other areas remains ambiguous to communities, which conflicts with the need for transparency [38]. Qualitative data from several studies reveals that in some circumstances communities are unaware of the roles and responsibilities of NGO staff, a key facet in building trust [34,38,39]. In addition, methods used for communicating information involve the establishment of committees that meet with community members at regular intervals [5], displaying information on news boards [28], social mapping [36], and informal propagation of information by word of mouth [9] that can promote transparency.

Despite regulations that demand financial transparency through submission of annual reports [40], in many circumstances, the sharing of information is voluntary and dependent on the willingness of NGOs to share data with communities [41]. Likewise, we found that quantifying social accountability remains a challenge, further limiting NGO transparency to communities [9]. Concepts that arose related to informal methods of information-sharing include knowledge gaps among individuals living in remote locations and those with limited contact, such as the elderly [11]. Conversely, in some studies the use of formal channels is perceived as impersonal, thereby creating distance

between NGOs and communities [35,39]. Moreover, claims of transparency are not always true, with as many as 25% of NGOs in one study declaring publicly accessible financial information that was ultimately not provided [39]. Furthermore, examples abound that reputable organizations may misrepresent information to uphold good standing at the expense of transparency [35,39].

### Program audit

Our study revealed that program audits, including self-regulation, play a crucial role in maintaining NGO credibility, legitimacy, and public trust (Table 3) [34]. We observed significant variations in implementation and impact within LMICs. In some cases, self-regulation improves transparency and resource management [39], while in others it may fail to prevent mismanagement, as seen with inflated program figures and financial mismanagement [42]. The effectiveness of self-regulation varies across NGOs. Some organizations leverage these mechanisms to enhance transparency and accountability, whereas others face challenges with enforcement and compliance [40]. The efficacy of these mechanisms can be compromised by power dynamics and the potential for misrepresentation [42]. Several examples indicate that the integrity of self-regulation could be compromised when field staff inflate scores to secure more funding, resulting in a loss of learning and information-sharing opportunities [8,42].

In addition, program audits deliver effective accountability through cultivating sustainable improvements in NGO performance and ethical practices in resource management [34]. A common theme found across studies is the prioritization of quantitative outcomes over process outcomes, which can undermine DA through oversimplified monitoring that inadequately captures the complexity of development programs [34,39]. This suggests a lack of effective assessment tools across the humanitarian sector, particularly for small NGOs [34]. In summary, while program audits are requisite to upward accountability, their role in delivering DA requires balanced internal and external oversight, with a commitment to transparency and quality improvement that aligns closely with beneficiary perspectives to be truly effective [34,35].

### Social audit

Feedback and community surveillance mechanisms offer a systematic approach to receiving perspectives from affected populations through social auditing (Table 3) [10]. This integrated mechanism ensures DA by monitoring service delivery and fostering trust in the NGO-beneficiary relationship [4]. While feedback systems can effectively amplify beneficiary voices, they often suffer from low engagement levels among beneficiaries [10]. Factors such as fear of losing aid, lack of rights awareness, and failure to act on feedback by NGOs influence the effectiveness of this mechanism [10]. Community-integrated dialogue with affected populations and building up rights-based awareness are aspects of social auditing found integral to the effective delivery of DA [4,10].

NGOs have experienced varying degrees of success in acquiring honest and critical feedback from beneficiaries [10,12,43]. Feedback channels that consider cultural norms and societal restrictions of beneficiaries were found to be vital in improving the usage of feedback systems, particularly regarding the inequitable representation of female participation [43]. Community-led processes demonstrate that stigma-free environments and privacy-focused behaviors were significant factors in the community's ability to provide feedback for effective DA [43]. The strength of feedback system implementation includes anonymity when reporting issues of corruption or fraud [42,43]. However, a notable weakness is the limited access to technical modes of delivery. We found instances where incomplete feedback loops hinder affected populations from feeling heard and being updated on how NGOs are acting on their feedback [1,42]. NGOs can believe they implemented feedback mechanisms according to their own definitions and goals, while in reality, it does not reach the community as they can have a different definition or expectation of feedback systems [1,42]. Our review suggests that improving communication within the NGO-beneficiary relationship promotes the effective delivery of DA in social auditing [42].

## Barriers to DA

Various studies demonstrated that implementation challenges exacerbate intricate macro-level barriers related to existing power asymmetries between donors and beneficiaries, and fragmentation within the humanitarian sector regarding the effective delivery of DA (Table 5) [12,44]. The obstacles to direct implementation strategies include logistical, practical, and cultural barriers that impact the sustainable outcomes of DA. Geographically and historically, organizations that fund aid are in a different region than the beneficiary community, creating a multifaceted disconnect that can inhibit DA [34,44]. Resource constraints create significant logistical barriers that limit staff recruitment, training, accurate data collection, and effective feedback mechanisms necessary for implementation [35]. The literature suggests that accountability measures increase as NGOs become larger [45,46]. The practical aspects of accountability can also be contingent on establishing clear guidelines and consistent practices within an organization's administration [1,28,47].

Power asymmetry is another significant issue, as we observed that management in some NGOs can have a low perception of the importance of DA. In some cases, there is explicit disbelief in the knowledge and skill capacity of beneficiaries to participate in their developmental aid [9,31,48]. Barriers that arise from cultural differences among stakeholders can influence the DA implementation process. Westernized individualistic approaches can create resistance from communities, requiring significant efforts to mitigate this barrier to gain productive input from beneficiaries [11,49].

Beneficiaries lack the social and financial currency that other NGO stakeholders have to enforce accountability [50,51]. This power imbalance necessitates DA, since it is not intrinsic but must be manufactured out of the goodwill of NGOs –a goodwill we ascribe to NGOs intrinsically, earned or not [18,52]. This element of "choice" in whether they participate in DA weakens its practice, as it has been consistently documented that beneficiary demands most often compete with and lose to the demands of more powerful stakeholders due to financial dependency or the array of direct barriers described [8,50]. Various studies indicate that power imbalance within communities leads to low representation of feedback from those fearing loss of aid and favors those with financial or social power, hindering effective DA by obstructing comprehensive participation [8,9,27,31,34]. This imbalance undermines the effectiveness of NGO interventions and erodes trust amongst beneficiaries, exacerbated by centralized decision-making that aligns more with government and private sector requirements than local needs [9,30,40]. This often results in NGOs failing to address grassroots issues and being accountable to their intended beneficiaries [12]. Relying on donor funds and limited local dialogue, we found that NGOs frequently import externally designed models rather than developing locally appropriate solutions [9,12].

**Table 5. Barriers to DA.**

| Implementation | | References |
|---|---|---|
| Logistical | NGO size in relation to governance practice, budget allocation, lack of organizational staff training, lack of resources | [10,28,40,50] |
| Practical | Lack of measures or impact monitoring, follow-up mechanisms, organizational commitment, action planning for local development, supply-driven initiative, lack of flexibility and adaptability, lack of social approaches | [9,11,12,28,47] |
| Cultural | Westernized application of DA, geographic disconnect | [11,12,55] |
| **Power asymmetry** | | |
| Structural systemic | Perceived beneficiary capability, historical colonization and decentralization of aid | [9,12,31,48] |
| Competing priorities | Lack of demand from beneficiaries to hold NGOs accountable, rights awareness, miscommunication | [9,27,31,34] |
| **Sectoral fragmentation** | | |
| Regulatory | Lack of sectoral governance structures and institutional strengthening, lack of universal guidelines and definitions, reliance on donor-driven initiative | [2,9,11,18] |
| Enforcement | Lack of legal protections, ability of NGOs to not consider themselves responsible | [8,40,53,54] |
| Sustainability | Lack of national level engagement from grassroots movement, negative staff attitudes and understanding, resource mismanagement | [8,9,32,38] |

An emerging theme intersecting these barriers is fragmentation across the NGO sector relating to DA. The lack of legal protection, universal guidelines, and standardized definitions for beneficiary accountability further limits its effective implementation [8,40]. The pervasive belief as indicated in the literature, that NGOs are not responsible for certain actions, exacerbates this issue [40,53]. Additionally, staff members who lack a clear understanding of DA due to the absence of standardized definitions and committed organizational direction, often develop poor attitudes towards accountability practices [1,8]. They may perceive efforts to improve accountability as punitive rather than constructive. Our study also observed that while the standardization and regulation of DA have the potential to influence power dynamics and fragmentation, these efforts may remain performative without genuine and effective implementation of DA practices [8,40,54].

## Discussion

Despite being central to NGO mission statements, DA remains underexplored in terms of its mechanisms, effects, and implementation measures. While much literature discusses barriers to DA, there is limited analysis of how NGOs address community feedback and engage in participatory evaluation, particularly regarding power asymmetry. Data suggests that informal accountability through participation often lacks depth, hindering effective accountability. Consequently, DA poses challenges to organizational legitimacy, as it remains unstandardized and subject to varied interpretations, which undermines genuine accountability to affected populations.

### Assessing the impact of DA mechanisms

The effectiveness of DA mechanisms can be evaluated based upon their impact on the sustainability of local development and organizational effectiveness (Table 6) [9,17]. Key indicators include the level of beneficiary influence in aid processes [18], the responsiveness and adaptability of programs to beneficiary need [28], the use of community-led key performance indicators [2], integration with existing local systems [28], the rate of completion feedback loops [2], and the enhancement of internal process capacity for effective implementation [45]. By incorporating both qualitative and quantitative metrics— such as stakeholder feedback, resource management outcomes, and programmatic adjustments— an evaluation tool would allow NGOs to comprehensively assess the effectiveness of their DA initiatives towards sustainable local development [9,17]. While beyond the scope of this paper, the measurement criteria outlined could form the basis of a future empirical tool to assess DA mechanisms.

### Cultural and social influence on DA mechanism effectiveness

The effectiveness of DA mechanisms in LMICs is significantly shaped by cultural and social dynamics [1,2]. Western accountability practices often fail to resonate in non-Western contexts, leading to systems that feel disconnected from local needs and values [8,11]. Despite this, the intersection of cross-cultural factors with DA has been insufficiently explored in existing literature [1,9].

Colonial legacies and historical power imbalances further complicate DA in LMICs. Communities that have long been subjected to externally driven development may view DA mechanisms as extensions of unequal power structures, fostering skepticism and disengagement. Decolonizing aid discussions emphasize reshaping power dynamics to empower local

**Table 6. Criteria for evaluating the effectiveness of DA mechanisms.**

| Criterion | | References |
|---|---|---|
| Answerability | The extent to which DA mechanisms enable organizations to respond to beneficiary feedback | [9,17,45] |
| Cultural fit | Alignment of aid processes with cultural norms and practices | [2,8,9,35] |
| Self-reliance | Degree to which communities achieve ownership and sustainability of development initiatives | [9,18,55] |
| Resource management | Efficiency in managing resources, minimizing waste, maximizing reach and penetration of aid activities | [12,18,28,45] |

communities in decision-making [1,35]. When DA mechanisms overlook these imbalances, they can inadvertently reinforce the inequalities they aim to address.

Our review suggests that NGOs often focus on technical accountability solutions, such as procedural transparency while neglecting the cultural and social factors that shape community engagement with DA mechanisms [1,2]. In many LMICs, social hierarchies and authority norms hinder participation, particularly for marginalized groups like women and minorities [1]. These groups often face significant barriers to having an equal voice in holding organizations accountable.

Tailoring DA mechanisms to local cultural and social contexts is essential. There is a growing call for mechanisms that prioritize local perspectives and decision-making, rather than imposing external standards. By doing so, DA mechanisms can become more relevant, effective, and empowering for the communities they are meant to serve.

## Enhancing organizational effectiveness and sustainability through DA

The ultimate goal of DA is to foster sustainable local development and enhance organizational effectiveness. The DA mechanisms explored in this paper offer multiple pathways to achieve these objectives, both as tools and direct processes that contribute to lasting impact. Mechanisms such as transparency, program audits, and social audits play a critical role in nurturing public trust by establishing open communication channels between NGOs and the communities they serve. Full disclosure includes providing beneficiaries with access to financial information, resource management, and project planning that empowers communities to make informed decisions and report any instances of misappropriation or unethical conduct. Feedback systems, especially those allowing for anonymity, further encourage active beneficiary participation in the DA process.

When implemented effectively, these DA mechanisms can help mitigate common challenges in aid delivery, such as failure to follow through or failure to act. A person-centered approach, as opposed to using standard assessment metrics, helps to build trust in the NGO-beneficiary relationship, reinforcing the legitimacy of the organization and facilitating demand-driven aid.

Addressing contextual needs through collaboration with local communities and authorities is essential for prioritizing local developmental aid. This can be achieved through beneficiary engagement at all stages of the aid process encompassing planning, decision-making, implementation, and monitoring. Active participation in these processes not only fosters self-sufficiency and empowerment but also ensures that interventions are culturally relevant and practically feasible.

The strength of DA mechanisms lies in their capacity to integrate transparency, social audits, and program audits within community engagement frameworks. This approach maximizes the return on investment in resource management, develops culturally appropriate interventions with high uptake, and ultimately strengthens organizational legitimacy and public trust. The overarching objectives of DA—beneficiary ownership and empowerment—serve as proxies for achieving organizational effectiveness and sustainability, contributing to the long-term success of development efforts.

## Improving DA in NGOs: Key factors and challenges

Quality leadership and positive attitudes of front-line staff are essential to improving the implementation of DA through training and fostering a supportive organizational culture [8,39]. NGOs can enhance DA by prioritizing beneficiary engagement, local leadership, and feedback mechanisms over excessive resources for formal evaluations [6,51]. Trust and cultural humility between NGOs and communities are crucial for promoting active participation and driving change [9,11].

However, NGOs face challenges integrating DA into their frameworks due to limited time, resources, and staff, often leading to over-reporting community involvement to preserve donor relationships [28,39]. DA is often treated as supplementary, with inadequate adaptation to beneficiary feedback, signaling a need for more organizational investment in DA [28]. The literature also supports making DA a requirement for donor funding [42,54].

Criticism of results-based management tools highlights their tendency to oversimplify social change and neglect local contexts, leading to rigid, one-size-fits-all solutions that hinder adaptability and responsiveness to community needs [8,52,55]. A balanced approach to DA is necessary, ensuring that affected populations actively shape outcomes rather than being passive recipients [11,12]. Shifting from the term "beneficiary" to recognizing communities as decision-makers can transform accountability dynamics and improve humanitarian aid effectiveness.

### Strengthening DA in humanitarian NGOs: A path forward

For effective DA in humanitarian NGOs, a systematic prioritization of local development through policy change is essential [28]. While initiatives like the Participation Revolution in the Grand Bargain aim to improve resource management, there is still much to be done in achieving demand-driven aid outcomes [56]. Policies should emphasize a rights-based approach, self-reliance, and integration with donor-driven DA to mitigate implementation barriers. Effective DA requires listening to stakeholders, including affected populations, and empowering them with equal voice and power [6,27].

The erosion of trust in humanitarian aid calls for policies founded on transparency, including regular communication with communities about project details and rights [8]. Standardized legal frameworks for engagement and resource management are also recommended [6,10,40]. Recent policy shifts favor bottom-up evaluation models, like the Grounded Accountability Model [57], which incorporates continuous community feedback to develop localized indicators. Although resource-intensive, such models yield high returns in community engagement.

DA and accountability to donors can complement each other. Engaging donors and governments directly with aid recipients cultivates trust and cooperation, ultimately enhancing NGO impact [54]. By viewing upward and DA as interconnected, NGOs can facilitate open communication, balance functional and strategic accountability, and shift focus from reporting to advancing sustainable, long-term transformation [1,9].

### Haiti: A case study on the need for DA in the "NGO Republic"

Haiti's history illustrates the consequences of inadequate DA mechanisms, particularly in post-crisis humanitarian efforts. Often referred to as "The NGO Republic," Haiti has seen extensive aid investment without significant tangible results, contributing to the country's ongoing crises of food insecurity, displacement, and gang violence [10,38]. Following the 2010 earthquake, NGOs flooded the country yet failed to meet the needs of the population. Three major DA barriers were identified: 1) Inappropriate allocation of resources, 2) Fragmented response and recovery efforts, and 3) Power asymmetry, where NGO priorities diverged from the needs of the population [10,38].

The influx of NGOs in post-earthquake Haiti displayed a lack of contextual understanding and direction, leading to activities that were misaligned with national health norms and abandoned without proper transition, which undermined health system recovery [10,38]. This lack of transparency and accountability led to distrust, leaving affected Haitians disempowered and uncertain about the future impact of aid. The failure to meet community needs underscores the demand for a shift toward strategic accountability, which emphasizes local leadership and equal participation to drive sustainable, effective delivery of DA [10,38].

### Research gaps in the effectiveness of DA

Our literature review reveals a significant research gap: a systematic review of DA mechanism effectiveness has never been conducted. This gap highlights the importance of DA in achieving positive outcomes in aid activities. While numerous studies discuss accountability mechanisms, there is a disproportionate focus on participatory approaches, with a lack of evaluation and reporting on DA practices (Table 3). Participation was cited 97 times, while program auditing was cited 69 times, yet evaluation reporting appeared only 6 times, indicating a lack of transparency and an oversimplification of DA as synonymous with participation.

Moreover, our review found no standardized criteria for measuring DA effectiveness. Table 3 shows that while the effectiveness of DA mechanisms is frequently discussed, the mechanisms themselves and their components are often overlooked. This gap stems from the historical uncertainty about what constitutes DA, making it challenging to evaluate

and measure its impact effectively. This gap in research emphasizes the need for clearer definitions, classifications, and methods to assess DA mechanisms and their outcomes.

## Strengths and limitations

A key theme across the literature is the bias in interviews regarding DA practices, particularly the lack of representation from beneficiaries' perspectives. Internal reports often overrepresent satisfaction with NGO activities due to fear of losing aid or feeling unqualified to offer negative feedback. Additionally, there is a gap in research related to the demographic representation of marginalized populations. The variability in data quality (S3 Table) further complicates the analysis, as some studies fail to account for interviewer bias or influence on participant responses. Moreover, the lack of detailed analytical methods makes it challenging to assess the rigor of studies across findings. However, triangulation and peer review were employed to enhance the qualitative review's rigor, utilizing diverse data collection methods, such as focus groups, surveys, and interviews. The strength of these methods lies in their potential applicability across various humanitarian contexts.

## Conclusion

This qualitative systematic review underscores the critical role of DA in enhancing organizational legitimacy, improving aid effectiveness, and fostering sustainable development outcomes. However, substantial research gaps remain regarding the effectiveness of DA mechanisms utilized by humanitarian NGOs within LMICs. This research deficit undermines the sustainability of local development efforts and, more broadly, diminishes organizational effectiveness. Complex barriers to implementing DA, such as power asymmetry and sectoral fragmentation, must be addressed through continuous assessment and innovative quality improvement methods. Implementing balanced accountability mechanisms that promote equality in power dynamics is essential for achieving meaningful and lasting outcomes for affected populations.

## Definitions

Downward accountability: being answerable to affected populations for actions, giving them influence over key decisions which are made throughout the project's lifetime that is a dynamic process of listening, adapting, and responding while creating sustainable systems. Accountability mechanisms: tools and processes that deliver accountability; tools can be measured at specific points in time with tangible documentation; processes are a course of action where the means of delivery is more tangible than the results. Social auditing: community surveillance, needs assessment, focus groups, surveys. Feedback Systems: complaint mechanisms, involvement of communities in evaluating activities, responsibility to improve quality of program actions. Program auditing: internal assessment, reporting and improving on social performance and ethical behavior, performance evaluation, self-regulation and enforcement. Evaluation reporting: transparency, disseminating information to communities on how finances are used, organizational structure, and program outcomes, communicating motives, resources, power dynamics, limitations of work, and decision making process, aims to maintain community trust, being upfront about access to human rights. Ownership: knowledge transfer, capacity building, sustainability for self-reliance, integration with existing systems, local leadership, beneficiaries making the decisions and those decisions are implemented. Participation: consulting, involving, collaborating, community engagement.

## Supporting information

**S1 Table. PRISMA checklist.**
(DOCX)

**S2 Table. Search strategy.**
(XLSX)

**S3 Table. Quality assessment.**

(XLSX)

## Author contributions

**Conceptualization:** Elizabeth Noble, Cathleen Seaman.

**Data curation:** Elizabeth Noble, Dina Moinul, Oumou Khairy Djim Sylla, Sophia Friedmann, Kristen Amick, Nehal Rowhani, Rashi Dua, Nowshin Mannan, Cathleen Seaman, Omobolanle Ayo, Shubhra Pant, Oluwatimilehin Osoko, Srija Gogineni, Chris Dickey, Emmanuel Peprah.

**Formal analysis:** Elizabeth Noble, Dina Moinul, Oumou Khairy Djim Sylla, Sophia Friedmann, Kristen Amick, Nehal Rowhani, Rashi Dua, Nowshin Mannan, Cathleen Seaman, Omobolanle Ayo, Shubhra Pant, Oluwatimilehin Osoko, Srija Gogineni, Chris Dickey, Emmanuel Peprah.

**Investigation:** Elizabeth Noble, Dina Moinul, Oumou Khairy Djim Sylla, Sophia Friedmann, Kristen Amick, Nehal Rowhani, Rashi Dua, Nowshin Mannan, Cathleen Seaman, Omobolanle Ayo, Shubhra Pant, Oluwatimilehin Osoko, Srija Gogineni, Carly Malburg, Chris Dickey, Emmanuel Peprah.

**Methodology:** Elizabeth Noble, Dina Moinul, Oumou Khairy Djim Sylla, Sophia Friedmann, Kristen Amick, Nehal Rowhani, Rashi Dua, Nowshin Mannan, Cathleen Seaman, Omobolanle Ayo, Shubhra Pant, Oluwatimilehin Osoko, Srija Gogineni, Carly Malburg, Chris Dickey, Emmanuel Peprah.

**Project administration:** Elizabeth Noble, Dina Moinul, Carly Malburg.

**Resources:** Elizabeth Noble, Dina Moinul, Oumou Khairy Djim Sylla, Sophia Friedmann, Kristen Amick, Nehal Rowhani, Rashi Dua, Nowshin Mannan, Cathleen Seaman, Omobolanle Ayo, Shubhra Pant, Oluwatimilehin Osoko, Srija Gogineni, Carly Malburg, Chris Dickey, Emmanuel Peprah.

**Software:** Elizabeth Noble, Dina Moinul, Oumou Khairy Djim Sylla, Sophia Friedmann, Kristen Amick, Nehal Rowhani, Rashi Dua, Nowshin Mannan, Cathleen Seaman, Omobolanle Ayo, Shubhra Pant, Oluwatimilehin Osoko, Srija Gogineni, Carly Malburg, Chris Dickey, Emmanuel Peprah.

**Supervision:** Elizabeth Noble, Dina Moinul, Carly Malburg, Emmanuel Peprah.

**Validation:** Elizabeth Noble, Dina Moinul, Oumou Khairy Djim Sylla, Sophia Friedmann, Kristen Amick, Nehal Rowhani, Rashi Dua, Nowshin Mannan, Cathleen Seaman, Omobolanle Ayo, Shubhra Pant, Oluwatimilehin Osoko, Srija Gogineni, Carly Malburg, Chris Dickey, Emmanuel Peprah.

**Visualization:** Elizabeth Noble, Dina Moinul, Oumou Khairy Djim Sylla, Sophia Friedmann, Kristen Amick, Nehal Rowhani, Rashi Dua, Nowshin Mannan, Cathleen Seaman, Omobolanle Ayo, Shubhra Pant, Oluwatimilehin Osoko, Srija Gogineni, Chris Dickey, Emmanuel Peprah.

**Writing – original draft:** Elizabeth Noble, Dina Moinul, Oumou Khairy Djim Sylla, Sophia Friedmann, Kristen Amick, Nehal Rowhani, Rashi Dua, Nowshin Mannan, Cathleen Seaman, Omobolanle Ayo, Shubhra Pant, Oluwatimilehin Osoko, Srija Gogineni, Chris Dickey, Emmanuel Peprah.

**Writing – review & editing:** Elizabeth Noble, Dina Moinul, Oumou Khairy Djim Sylla, Sophia Friedmann, Kristen Amick, Nehal Rowhani, Rashi Dua, Nowshin Mannan, Cathleen Seaman, Omobolanle Ayo, Shubhra Pant, Oluwatimilehin Osoko, Srija Gogineni, Carly Malburg, Chris Dickey, Emmanuel Peprah.

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
