## [Decision Letter · Decision Letter 0]

11 Sep 2024

PONE-D-24-30081Downward accountability mechanism effectiveness by non-governmental organizations in low- and middle-income countries: A qualitative systematic reviewPLOS ONE

Dear Dr. Noble,

Thank you for submitting your manuscript to PLOS ONE. After careful consideration, we feel that it has merit but does not fully meet PLOS ONE’s publication criteria as it currently stands. Therefore, we invite you to submit a revised version of the manuscript that addresses the points raised during the review process.

We look forward to receiving your revised manuscript.

Kind regards,

Asif Khan

Guest Editor

PLOS ONE

2. We note that there is identifying data in the Supporting Information file < S2 Quality Assessment.xlsx>. Due to the inclusion of these potentially identifying data, we have removed this file from your file inventory. Prior to sharing human research participant data, authors should consult with an ethics committee to ensure data are shared in accordance with participant consent and all applicable local laws.

-Location data

3. As required by our policy on Data Availability, please ensure your manuscript or supplementary information includes the following:

Additional Editor Comments:

Dear Author(s),

The manuscript has been reviewed by the reviewers, as per thier suggestions/comments, I shall offer you another chance to revise the manuscript and per their suggestions.

Kindly revise the manuscript as per the comments, submit revised version of the manuscript (clearly highlight the changes) and response sheet.

Regards

Dr. Asif Khan

Reviewers' comments:

Reviewer's Responses to Questions

**Comments to the Author**

1. Is the manuscript technically sound, and do the data support the conclusions?

Reviewer #1: Partly

Reviewer #2: Yes

2. Has the statistical analysis been performed appropriately and rigorously? 

Reviewer #1: Yes

Reviewer #2: Yes

3. Have the authors made all data underlying the findings in their manuscript fully available?

Reviewer #1: Yes

Reviewer #2: No

4. Is the manuscript presented in an intelligible fashion and written in standard English?

Reviewer #1: No

Reviewer #2: Yes

5. Review Comments to the Author

Reviewer #1: Word documents attached for the comments, I would request the authors to address all the comments for better clarity. Authors need to work on the gap formulation with the recommendation part and also to re-check grammar is also important.

Reviewer #2: The paper is Good, however, require many changes

Comments

1. Revise introduction section properly, incorporate latest study of the area. Clearly communicate the research contribution and your novelty. And mention how your study is different from other.

2. In the literature/hypothesis development; The author(s) need to critically examine the previous study and offer a comprehensive literature in your domain.

3. Why you have opted for this methodology, do you have any other alternative methodology, mention your justifications of selecting the tool, variables.

4. The discussion can be more clear and link it with the current literature. Justify your finding and analysis in more logical and clear manner.

5. Policy Implications and limitations of the study

6. Check for any typo and grammar errors.

6. PLOS authors have the option to publish the peer review history of their article (what does this mean? ). If published, this will include your full peer review and any attached files.

**Do you want your identity to be public for this peer review?** For information about this choice, including consent withdrawal, please see our Privacy Policy .

Reviewer #1: **Yes: ** SHAKEB AKHTAR

Reviewer #2: No

---

## [Author Response · Author response to Decision Letter 1]

7 Dec 2024

Editor Comment 1: Please ensure that your manuscript meets PLOS ONE's style requirements, including those for file naming.

Author Response to Editor Comment 1: Thank you for this comment. We appreciate your thorough review of our paper. We can confirm our paper adheres to the journal's style requirements and guidelines.

Editor Comment 2: We note that there is identifying data in the Supporting Information file < S2 Quality Assessment.xlsx>. Due to the inclusion of these potentially identifying data, we have removed this file from your file inventory. Prior to sharing human research participant data, authors should consult with an ethics committee to ensure data are shared in accordance with participant consent and all applicable local laws.

Author Response to Editor Comment 2: We appreciate the reviewer’s thorough review of our manuscript. Our systematic review does not contain any personally identifiable information (PII) because we solely relied on publicly available literature and extracted data from those sources.

Editor Comment 3: As required by our policy on Data Availability, please ensure your manuscript or supplementary information includes the following: A numbered table of all studies identified in the literature search, including those that were excluded from the analyses.

Confirmation that the study was eligible to be included in the review

Name of extractors and date of extraction can be added to the methods section.

All data extracted from each study for the reported systematic review and/or meta-analysis that would be needed to replicate your analyses

Author Response to Editor Comment 3: We thank the reviewer for this valuable suggestion. To enhance the transparency of our methodology, we have uploaded the included and excluded studies to Open Science Framework (OSF). Additionally, we have added a clarification in the manuscript to ensure readers understand that the dataset is publicly accessible. The dataset, which contains over 1,000 publications, can be accessed here (https://doi.org/10.17605/OSF.IO/3JNMA). We have also included our search criteria, strategy, and extraction framework table as supplementary materials in the re-submission of this manuscript. We have also updated the language in our method section to clearly describe how we screened, excluded, and included our selected articles. We feel these changes align with the data availability policy, enhances the transparency of our work, and will allow our work to be replicated for future analyses.

Reviewer 1 Comment 1: Is the discussion of the significant research gaps in the effectiveness of downward accountability (DA) mechanisms among NGOs in LMICs comprehensive and supported by sufficient evidence? There is still a lacking behind the research gap foundation. I would advise the authors to rework on it.

Author Response to Reviewer 1 Comment 1: Thank you for your insightful feedback. In response to your comment, we have restructured and revised the Discussion section to provide a more comprehensive and evidence-based analysis of the significant research gaps in the effectiveness of DA mechanisms amongst NGOs in LMICs. Please refer to the updated text on page 35, lines 555–568, where we have incorporated a clearer synthesis of the existing evidence and explicitly highlighted the foundational gaps in the current research. We believe these revisions strengthen the discussion and provide a more robust foundation for understanding the gaps in this critical area. We appreciate the reviewer’s suggestion to improve this aspect of the manuscript.

Reviewer 1 Comment 2: What concerns has been addressed about the transparency of data presented in the article, particularly regarding the selection and exclusion criteria used during the systematic review process? The authors need to identify and explain them in detail.

Author Response to Reviewer 1 Comment 2: Thank you for your valuable comment. We fully appreciate the importance of transparency, particularly in the systematic review process. In response to your concern, we have taken several steps to ensure that the selection and exclusion criteria, as well as the overall methodology, are clearly communicated and accessible to the readers.

We have included a detailed table outlining our search criteria, strategy, and data extraction framework as supplementary materials in the re-submission of the manuscript. This table provides a transparent overview of the criteria we used to identify relevant studies and how we handled the data throughout the review process. Additionally, we have updated the Methods section of the manuscript to clarify the screening, inclusion, and exclusion procedures used to select the studies, with a focus on providing greater detail regarding each step. Please refer to the revised text on page 8, lines 170–175, for these updates.

To further enhance transparency, we have made all the data from our review publicly available through the Open Science Framework (OSF). This includes our full methodology, such as the PRISMA flowchart, as well as the results of the review. By providing open access to this data, we aim to allow other researchers to scrutinize and replicate our process, ensuring the highest standards for transparency and reproducibility. We are committed to maintaining an open and transparent approach, and we believe this fulfills the expectations for rigorous and trustworthy data reporting in peer-reviewed publications.

Reviewer 1 Comment 3: The themes and codes derived from the qualitative analysis are not clearly defined and consistently applied throughout the article.

Author Response to Reviewer 1 Comment 3: Thank you for your valuable feedback. In response to your comment, we have clarified our thematic analytical approach by adding detailed language on page 9, lines 193–207. This section now explains how we derived themes and codes for our qualitative analysis. Additionally, we have thoroughly reviewed the manuscript to ensure these themes and codes are consistently applied across the results, tables, and discussion sections. We believe these updates enhance the coherence and rigor of our analysis. We sincerely thank the reviewer for this constructive suggestion.

Reviewer 1 Comment 4: The article does not adequately address the practical barriers to implementing DA mechanisms in NGOs. And the recommendations provided to overcome these challenges are not also properly explained with the right implication to it.

Author Response to Reviewer 1 Comment 4: Thank you for your insightful comment. We appreciate your thorough review of our paper. Our primary objective was to assess the existing literature on DA and identify the challenges reported within it. In response to your feedback, we have expanded the discussion on pages 33–34, lines 498–536, to provide a more detailed analysis of the practical barriers to implementing DA mechanisms in NGOs. Additionally, we have clarified and elaborated on the recommendations for overcoming these challenges, ensuring they are better aligned with the implications drawn from the current literature. We hope these enhancements provide greater clarity and value to this important aspect of our study.

Reviewer 1 Comment 5: There needs to be a balanced discussion on both the strengths and weaknesses of the various accountability mechanisms, such as participation, ownership, and social auditing, which I felt is missing in the paper.

Author Response to Reviewer 1 Comment 5: We appreciate your thorough review of our paper. Please refer to the highlighted text on page 31-32, line numbers 468-496; here we have expanded on the strengths and weaknesses of the different DA mechanisms, offering a more balanced discussion of these concepts to our audience.

Reviewer 1 Comment 6: The cultural and social factors that have been included in the research for influencing the effectiveness of DA mechanisms, needs to be sufficiently explored, particularly in the context of different LMICs.

Author Response to Reviewer 1 Comment 6: Thank you for your thoughtful feedback. In response to your comment, we have added a new section on pages 30–31, lines 445–466 within the Results section, focusing on how cultural and social factors can influence the effectiveness of DA mechanisms. This addition provides greater insight into these important contextual factors, particularly in the context of various LMICs. However, we acknowledge that a comprehensive exploration of cultural and social influences is beyond the scope of this systematic review. We believe such an analysis requires a more in-depth qualitative approach, and a separate manuscript addressing these aspects is currently under consideration. We appreciate the reviewer for highlighting this important direction for future research.

Reviewer 1 Comment 7: How does the article provide clear criteria for assessing the impact of DA mechanisms on the sustainability of local development initiatives and organizational effectiveness? This question needs to be answered for more clarity.

Author Response to Reviewer 1 Comment 7: We appreciate the reviewer’s thorough assessment of our manuscript. In response, we have expanded our discussion on page 30, lines 431–442, and in Table 6, where we elaborate on how we assessed the impact of DA mechanisms on aid activities and overall NGO effectiveness. We recognize the importance of establishing clear criteria for evaluating the impact of DA mechanisms on sustainability. However, a lengthy discussion on this topic extends beyond the scope of this qualitative systematic review. We appreciate the opportunity to highlight this as a critical area for future research.

Reviewer 2 Comment 1: The paper presents an analytical look at a very interesting policy question. The paper needs to be better organized with major modifications. Revise the introduction section properly, incorporating the latest study of the area. Clearly communicate the research contribution and your novelty. And mention how your study is different from others.

Author Response to Reviewer 2 Comment 1: In response to your comment, we have extensively revised and reorganized the Introduction section to better structure the paper and enhance its clarity. These revisions incorporate the latest studies in the field, ensuring that our work is contextualized within the current state of knowledge. Additionally, we have clearly articulated the novelty of our research, emphasizing how it contributes uniquely to addressing the identified policy question. We have also explicitly highlighted how our study differs from and builds upon previous work. Please refer to the updated text on pages 4–7, particularly lines 131–141, where these improvements are detailed.

Reviewer 2 Comment 2: In the literature/hypothesis development; The author(s) need to critically examine the previous study and offer a comprehensive literature in your domain.

Author Response to Reviewer 2 Comment 2: Thank you for your thoughtful feedback. In response to your comment, we have critically examined the existing literature and significantly expanded the Introduction section to provide a more comprehensive background on the current state of research in our domain. We also rerun the search in November 2024 to find any new articles meeting our search criteria, as our search was completed in December 2023, and determined that no new publications met our inclusion criteria. These additions offer a deeper analysis of previous studies and highlight key gaps that our study aims to address. Furthermore, we have used this enriched context to refine our hypothesis and clearly articulate the purpose of our research. The specific changes can be found on pages 5–6, lines 105–116, where we have highlighted the new content added in response to your suggestion.

Reviewer 2 Comment 3: Why have you opted for this methodology, do you have any other alternative methodology, mention your justifications of selecting the tool, variables, methods.

Author Response to Reviewer 2 Comment 3: To address your query, we chose a systematic review methodology because it is the most rigorous approach for synthesizing existing evidence to answer our research question. While alternative methodologies such as scoping or umbrella reviews exist, the systematic review was deemed appropriate for this study due to the significant gap in the literature on DA. Our methodology was carefully designed in collaboration with a qualified medical librarian, ensuring a comprehensive and reproducible search strategy. This approach allowed us to thoroughly evaluate the current state of the literature, addressing both the breadth and depth of the topic. To justify our methodology, we have provided additional details on page 7, lines 146–149, highlighting our rationale for selecting this approach, along with the search terms, inclusion/exclusion criteria, and data extraction variables. Furthermore, the complete search strategy has been included in Supplementary Materials S2 for transparency and reproducibility.

Reviewer 2 Comment 4: The discussion can be more clear and link it with the current literature. Justify your finding and analysis in a more logical and clear manner.

Author Response to Reviewer 2 Comment 4: Thank you for your insightful feedback. In response to your comment, we have significantly revised and reorganized the discussion section to ensure that our findings are clearly and logically linked to the current literature. Specifically, we have provided more detailed justifications for our findings by integrating them with relevant studies, highlighting both consistencies and discrepancies. This approach not only strengthens the logical flow, but also provides a clearer context for interpreting our results within the broader body of evidence. We believe these changes enhance the clarity and depth of the discussion, ensuring it aligns with the expectations of the scholarly community.

Reviewer 2 Comment 5: Policy Implications and Limitations/Challenges of policy

Author Response to Reviewer 2 Comment 5: In response, we have revised and expanded the section on policy implications in the Discussion. These revisions provide a more comprehensive and detailed outline of the policy implications for DA, with a focus on enhancing clarity and demonstrating the practical impact of our findings. Additionally, we have addressed the limitations and challenges associated with these policy implications, ensuring a balanced perspective. The updated section can be found on pages 33–34, lines 517–536.

Reviewer 2 Comment 6: Check for any typo and grammar errors

Author Response to Reviewer 2 Comment 6: We appreciate the reviewer’s thorough review of our manuscript. We have taken this opportunity to address all typographical or grammatical errors.

---

## [Decision Letter · Decision Letter 1]

3 Jan 2025

PONE-D-24-30081R1Downward accountability mechanism effectiveness by non-governmental organizations in low- and middle-income countries: A qualitative systematic reviewPLOS ONE

Dear Dr. Noble,

Thank you for submitting your manuscript to PLOS ONE. After careful consideration, we feel that it has merit but does not fully meet PLOS ONE’s publication criteria as it currently stands. Therefore, we invite you to submit a revised version of the manuscript that addresses the points raised during the review process.

We look forward to receiving your revised manuscript.

Kind regards,

Praveen Suthar, MPH

Academic Editor

PLOS ONE

Reviewers' comments:

Reviewer's Responses to Questions

**Comments to the Author**

1. If the authors have adequately addressed your comments raised in a previous round of review and you feel that this manuscript is now acceptable for publication, you may indicate that here to bypass the “Comments to the Author” section, enter your conflict of interest statement in the “Confidential to Editor” section, and submit your "Accept" recommendation.

Reviewer #1: All comments have been addressed

2. Is the manuscript technically sound, and do the data support the conclusions?

Reviewer #1: Yes

3. Has the statistical analysis been performed appropriately and rigorously? 

Reviewer #1: Yes

4. Have the authors made all data underlying the findings in their manuscript fully available?

Reviewer #1: Yes

5. Is the manuscript presented in an intelligible fashion and written in standard English?

Reviewer #1: Yes

6. Review Comments to the Author

Reviewer #1: All comments has been addressed. But the manuscript needs a proof read with a native english person before publishing the final version.

7. PLOS authors have the option to publish the peer review history of their article (what does this mean? ). If published, this will include your full peer review and any attached files.

**Do you want your identity to be public for this peer review?** For information about this choice, including consent withdrawal, please see our Privacy Policy .

Reviewer #1: No

---

## [Author Response · Author response to Decision Letter 2]

15 Jan 2025

Reviewer 1 Comment 1: All comments has been addressed. But the manuscript needs a proofread with a native english person before publishing the final version.

Author Response to Reviewer 1 Comment 1: The authors have thoroughly reviewed the manuscript and addressed the suggested edits as indicated by the reviewer. We thank the reviewer for their careful review and believe we have identified and corrected all minor proofreading errors.

---

## [Decision Letter · Decision Letter 2]

22 Apr 2025

Downward accountability mechanism effectiveness by non-governmental organizations in low- and middle-income countries: A qualitative systematic review

PONE-D-24-30081R2

Dear Dr. Noble,

We’re pleased to inform you that your manuscript has been judged scientifically suitable for publication and will be formally accepted for publication once it meets all outstanding technical requirements.

Kind regards,

Ali B. Mahmoud, Ph.D.

Academic Editor

PLOS ONE

Additional Editor Comments (optional):

Reviewers' comments:

Reviewer's Responses to Questions

**Comments to the Author**

1. If the authors have adequately addressed your comments raised in a previous round of review and you feel that this manuscript is now acceptable for publication, you may indicate that here to bypass the “Comments to the Author” section, enter your conflict of interest statement in the “Confidential to Editor” section, and submit your "Accept" recommendation.

Reviewer #3: All comments have been addressed

2. Is the manuscript technically sound, and do the data support the conclusions?

Reviewer #3: Yes

3. Has the statistical analysis been performed appropriately and rigorously? 

Reviewer #3: N/A

4. Have the authors made all data underlying the findings in their manuscript fully available?

Reviewer #3: Yes

5. Is the manuscript presented in an intelligible fashion and written in standard English?

Reviewer #3: Yes

6. Review Comments to the Author

Reviewer #3: The authors addressed all comments from reviewers. The manuscript reads well and much stronger.

well-done authors

7. PLOS authors have the option to publish the peer review history of their article (what does this mean? ). If published, this will include your full peer review and any attached files.

**Do you want your identity to be public for this peer review?** For information about this choice, including consent withdrawal, please see our Privacy Policy .

Reviewer #3: **Yes: ** Eve Namisango

---

## [Editor Report · Acceptance letter]

PONE-D-24-30081R2

PLOS ONE

Dear Dr. Noble,

I'm pleased to inform you that your manuscript has been deemed suitable for publication in PLOS ONE. Congratulations! Your manuscript is now being handed over to our production team.

Kind regards,

on behalf of

Dr. Ali B. Mahmoud

Academic Editor

PLOS ONE